# Estimating the health impact of nicotine exposure by dissecting the effects of nicotine versus non-nicotine constituents of tobacco smoke: A multivariable Mendelian randomisation study

Jasmine N. Khouja[1,2]*, Eleanor Sanderson[2,3], Robyn E. Wootton[1,2,4], Amy E. Taylor[2,3], Billy A. Church[5], Rebecca C. Richmond[2,3], Marcus R. Munafò[1,2,6]

1 School of Psychological Science, University of Bristol, Bristol, United Kingdom, 2 Medical Research Council Integrative Epidemiology Unit, University of Bristol, Bristol, United Kingdom, 3 Population Health Sciences, Bristol Medical School, University of Bristol, Bristol, United Kingdom, 4 Nic Waals Institute, Lovisenberg diakonale sykehus, Oslo, Norway, 5 School of Psychology and Vision Sciences, University of Leicester, Leicester, United Kingdom, 6 NIHR Bristol Biomedical Research Centre, Bristol, United Kingdom

* jasmine.khouja@bristol.ac.uk

**Data Availability Statement:** Data required to replicate all study findings reported in the article

## Abstract

The detrimental health effects of smoking are well-known, but the impact of regular nicotine use without exposure to the other constituents of tobacco is less clear. Given the increasing daily use of alternative nicotine delivery systems, such as e-cigarettes, it is increasingly important to understand and separate the effects of nicotine use from the impact of tobacco smoke exposure. Using a multivariable Mendelian randomisation framework, we explored the direct effects of nicotine compared with the non-nicotine constituents of tobacco smoke on health outcomes (lung cancer, chronic obstructive pulmonary disease [COPD], forced expiratory volume in one second [FEV-1], forced vital capacity [FVC], coronary heart disease [CHD], and heart rate [HR]). We used Genome-Wide Association Study (GWAS) summary statistics from Buchwald and colleagues, the GWAS and Sequencing Consortium of Alcohol and Nicotine, the International Lung Cancer Consortium, and UK Biobank. Increased nicotine metabolism increased the risk of COPD, lung cancer, and lung function in the univariable analysis. However, when accounting for smoking heaviness in the multivariable analysis, we found that increased nicotine metabolite ratio (indicative of decreased nicotine exposure per cigarette smoked) decreases heart rate (b = -0.30, 95% CI -0.50 to -0.10) and lung function (b = -33.33, 95% CI -41.76 to -24.90). There was no clear evidence of an effect on the remaining outcomes. The results suggest that these smoking-related outcomes are not due to nicotine exposure but are caused by the other components of tobacco smoke; however, there are multiple potential sources of bias, and the results should be triangulated using evidence from a range of methodologies.

are provided on Github at https://doi.org/10.5281/zenodo.10469666. Additionally, all numerical data for plots included in the manuscript can be found in the supplementary materials. All summary statistics can be found in the supplementary material.

**Funding:** This work is supported by the Medical Research Council (MRC) Integrative Epidemiology Unit at the University of Bristol (MC_UU_00032/01, MC_UU_00032/04, MC_UU_00032/07). The salary of REW is funded by the Norwegian South-Eastern Regional Health Authority (2020024). This work was supported by Cancer Research UK (grant number C18281/A29019) who fund the salary of JNK and RCR. The salary of AET is funded by the European Research Council Advanced Grant (ART-Health; Grant ERC number 101021566). This work is supported by the National Institute for Health Research (NIHR) Biomedical Research Centre at University Hospitals Bristol NHS Foundation Trust and the University of Bristol. The views expressed in this publication are those of the authors and not necessarily those of the NHS, the NIHR or the Department of Health and Social Care. The funders had no role in study design, data collection and analysis, decision to publish, or preparation of the manuscript.

**Competing interests:** The authors have declared that no competing interests exist.

## Author summary

Although we know that smoking tobacco negatively impacts health, we know relatively little about whether nicotine plays a role in causing poor health outcomes. When used for short periods of time, nicotine appears to have little impact on health. However, until recently, nicotine has rarely been used for long periods without accompanying exposure to tobacco smoke, so it is hard to disentangle the effects of regular nicotine use from the effects of tobacco smoke exposure. In this study, we aimed to dissect the effects of nicotine versus non-nicotine constituents of tobacco smoke on heart and lung health using multivariable Mendelian randomisation and data from a range of sources (including UK Biobank). We found that nicotine does not appear to be an independent cause of poor lung function, lung cancer, chronic obstructive pulmonary disease, or coronary heart disease, but does increase heart rate. These results support previous evidence which suggests that nicotine on its own does not directly cause poor health outcomes, with the exception of increasing heart rate.

## Introduction

Although the detrimental health effects of smoking are well known [1], it is less clear which individual components of cigarette smoke–such as nicotine–drive these effects. In recent years, it has become increasingly important to fill this knowledge gap, given the increase in the popularity of alternative nicotine delivery systems, in particular e-cigarettes [2]. E-cigarettes are often used by smokers to stop smoking and are generally used regularly for longer periods than conventional cessation aids which also contain nicotine (e.g., nicotine patches) [3,4]. Although *short-term* use of nicotine without the remaining constituents of tobacco smoke is relatively harmless, the harms of regular nicotine use are poorly understood [5]. Given many smokers choose to use e-cigarettes to reduce their risk of poor health outcomes [2], and that e-cigarettes can be used with or without nicotine, smokers and e-cigarette users need causal evidence regarding the possible health outcomes related to regular nicotine use to make informed decisions regarding their e-cigarette use.

Randomised controlled trials (RCT) are often employed to explore causal relationships [6], but an RCT would be unethical and impractical in this scenario–non-smokers would need to be unnecessarily exposed to nicotine for decades to understand the (potentially harmful) long-term effects of regular nicotine use without confounding from exposure to cigarette smoke. Mendelian randomisation (MR) is an alternative method which can be used in scenarios where RCTs are implausible or unethical. MR uses genetic variants associated with the exposure of interest to serve as instrumental variables to estimate the effect of that exposure on a particular outcome [7,8]. When using time-varying exposures in MR (e.g., smoking heaviness), MR results are interpreted as the effect of changing the liability of exposure at the time point measured [9,10]. Assuming the genetic variants have a proportionally constant effect on the exposure across the lifetime, the MR effect estimate will be the genetically predicted lifetime effect of being a heavier smoker across the life course [11,12]. Nonetheless, they require well powered genome-wide association studies (GWAS) to identify such genetic variants. There are currently no available GWAS of e-cigarette use or nicotine exposure without exposure to tobacco. Furthermore, there are insufficient data available to conduct a well-powered GWAS. Novel methods are therefore required to explore this research question.

Multivariable MR (MVMR) is an extension of MR that can be used to explore the direct effect of one exposure on an outcome while accounting for the effect of another (potentially

correlated) exposure or exposures [13]. The method is robust to pleiotropic effects of the single nucleotide polymorphisms (SNPs) used as instruments through the other exposures included in the model. As large-scale GWAS have identified SNPs that are associated with smoking heaviness [14] (i.e., the number of cigarettes a person smokes per day) and the nicotine metabolite ratio (i.e., how quickly a person metabolises nicotine–see below) [15] among smokers, we can employ this method to explore the effect of nicotine on selected health outcomes. Rather than exploring the total effect of smoking on a health outcome (which includes the effect of nicotine), this method allows us to separate the direct effect of nicotine versus the other constituents of tobacco smoke.

However, direct measurement of nicotine is difficult. Metabolites of nicotine such as cotinine (a direct metabolite) and 3'hydroxycotinine (3HC; a metabolite of cotinine) are often used as objective measures to proxy for nicotine given the short half-life of nicotine itself [16] (S1 Fig). Cotinine and cotinine plus 3HC are highly specific biomarkers, but both are impacted by metabolism (which can differ between individuals). The nicotine metabolite ratio (NMR), is a measure of how quickly a person metabolises nicotine (3'hydroxycotinine/cotinine) and therefore causally impacts the amount of nicotine in a person's body given a set amount of nicotine exposure [17]. Fig 1 illustrates that a smoker with a higher NMR (Person A) will have less circulating nicotine in their body than a smoker with a lower NMR (Person B) given the same

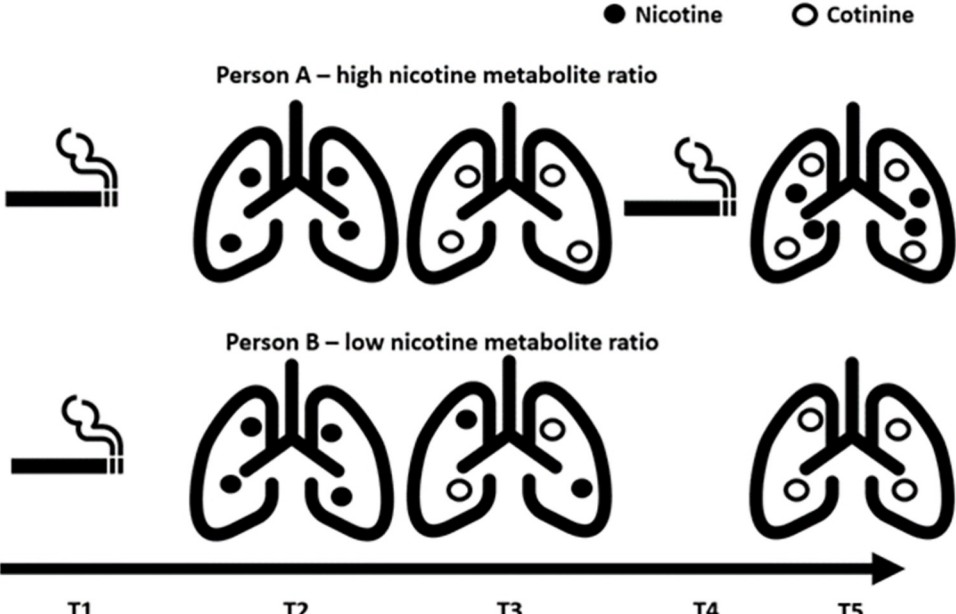

**Fig 1. Illustration of the impact of the nicotine metabolite ratio on circulating nicotine and smoking heaviness.**
Note: This illustration shows the differences in nicotine exposure between two people who smoke: Person A who has a high nicotine metabolite ratio, and Person B who has a low nicotine metabolite ratio. At timepoint 1 (T1) both Person A and B smoke one cigarette and inhale the same amount of nicotine which can be seen circulating in their body at timepoint 2 (T2). Later, at timepoint 3 (T3), Person A will have less circulating nicotine in their body than Person B (despite having inhaled the same level of nicotine) as more of the nicotine has been metabolised into cotinine. However, because smokers with higher nicotine metabolite ratios clear nicotine more quickly from their system, this often results in them smoking more cigarettes per day. So, at timepoint 4 (T4), Person A smokes another cigarette whereas Person B does not. This results in Person A having more circulating nicotine than Person B over the same time period [18,19]. This figure includes images from the following sources in accordance with their copyright licence: https://commons.wikimedia.org/wiki/File:Bootstrap_lungs.svg; https://commons.wikimedia.org/wiki/File:Noun_146. svg. The lung images have been adapted to include circles.

level of nicotine exposure; however, because smokers with a higher NMR clear nicotine quickly from their system, this often results in them smoking more cigarettes per day (CPD) [18,19]. In a standard MR model with NMR as an exposure, the result may be ambiguous as it could reflect the effect of higher nicotine exposure resulting from increased smoking heaviness or lower nicotine exposure resulting from faster metabolism. In contrast, when both NMR and CPD are included in an MVMR model (Fig 2), the analysis is analogous to a fixed access experiment in which the amount of smoke exposure / cigarettes smoked is fixed and the effect of nicotine exposure per cigarette smoked can be assessed. Therefore, in an MVMR framework accounting for genetic predisposition to CPD, the results should dissect the direct effect of nicotine levels per cigarette smoked from the direct effect of the other constituents of tobacco smoke, whereby an increase in NMR reflects lower exposure to nicotine (Fig 1).

Using this framework, it is possible to explore the effects of nicotine and the effects of the remaining constituents of tobacco smoke on health outcomes known to be impacted by smoking, and which therefore may be impacted by regular nicotine use, such as cancer and heart and lung function [20–22]. We employed a two-sample MVMR framework to explore the direct effects of nicotine compared with the non-nicotine constituents of tobacco smoke (using genetic proxies for NMR and for CPD) on health outcomes (lung cancer, chronic obstructive pulmonary disease [COPD], forced expiratory volume in one second [FEV-1], forced vital capacity [FVC], coronary heart disease [CHD] and heart rate [HR]). These outcomes were selected as previous evidence has indicated they are causally impacted by smoking heaviness [1,20–24]. The outcomes also include positive and negative control outcomes to aid causal inference; nicotine use is known to increase heart rate, but is not thought to be carcinogenic, so if the results do not provide evidence of an effect of nicotine on heart rate, and / or if they suggest an effect of nicotine on lung cancer, the results may indicate bias from a violation of the assumptions of MR [25–28]. In supplemental analyses, our secondary aim is to compare the results with analyses using alternative proxies for nicotine exposure (i.e., cotinine and cotinine plus 3HC).

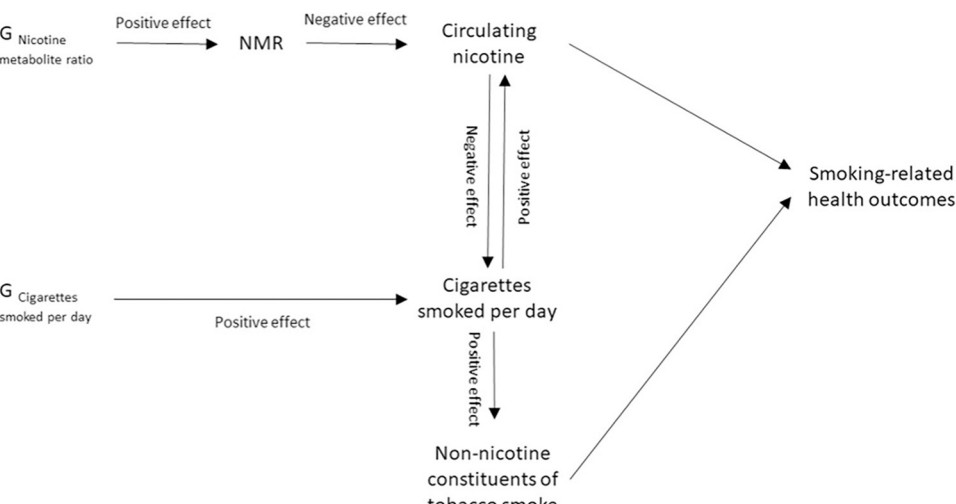

**Fig 2. Schematic of the study model.** Note: This schematic shows the proposed causal pathways of the study model. Where the causal directions between variables are known (i.e., evidenced in previous research), the direction of the effect is indicated as 'positive' or 'negative'. NMR = Nicotine Metabolite Ratio. G = genetic variants associated with the named exposure.

**Table 1. Descriptive statistics for participants included in the genome-wide association studies of lung cancer, coronary heart disease, chronic obstructive pulmonary disease, lung function and heart rate.**

| | Ever smokers | Current Smokers | Former Smokers | Never Smokers |
|---|---|---|---|---|
| | International Lung Cancer Consortium | | | |
| N | 40,187 | N/A | N/A | 9,859 |
| Lung Cancer (case rate) | 57% | N/A | N/A | 24% |
| | UK Biobank | | | |
| N | 213,341 | 49,721 | 163,620 | 258,056 |
| Coronary Heart Disease (case rate) | 11% | 11% | 11% | 6% |
| Chronic Obstructive Pulmonary Disease (case rate) | 3% | 6% | 2% | <1% |
| Forced Expiratory Volume (mean litres [SD]) | 2.82 (0.78) | 2.78 (0.83) | 2.83 (0.76) | 2.88 (0.78) |
| Forced Vital Capacity (mean litres [SD]) | 3.78 (0.97) | 3.81 (1.02) | 3.78 (0.96) | 3.78 (0.99) |
| Heart Rate (mean beats per minute [SD]) | 69.05 (11.38) | 71.27 (11.58) | 68.40 (11.23) | 68.93 (11.10) |

Note: SD = standard deviation

## Results

### Descriptive statistics

Smoking heaviness data were collected from 337,334 current and former smokers who reported their current or past smoking behaviour respectively (analyses corrected for current versus former status, average bin 3 [SD = 1] equating to 16–25 cigarettes per day) [14]. NMR data were collected from 5,185 current smokers [15]. Case rates and means (with standard deviations) for the outcomes in each of the samples included in the analyses are shown in Table 1.

### Instrument strength and heterogeneity

Where the Cochran's Q statistic (reported S1–S6 Tables) was greater than the number of SNPs included in the model, it is advised to focus on pleiotropy robust methods as this indicates heterogeneity and potential pleiotropy. MR-Egger and MVMR-Egger give estimates that are robust to directional horizontal pleiotropy under the assumption that this pleiotropy is uncorrelated with the strength of association between the SNP and the exposure [29]. However, using the MR-Egger and MVMR-Egger methods limits the statistical power of the analysis compared to inverse variance weighted (IVW) methods. Therefore, although we present the IVW results in the text, we have explicitly stated where the sensitivity analyses differ substantially (i.e., where the results could lead to different conclusions) and in these cases, additionally compare the Egger results (and weighted mode and weighted median results) in the MR analyses.

The F-statistics indicate that the SNPs included in these analyses are strong instruments for assessing the direct effects of smoking heaviness while accounting for the effect of the NMR (Fs = 33.96 and 34.17) and for assessing the direct effects of the NMR while accounting for smoking heaviness (Fs = 30.17 and 49.08).

### Binary outcomes

The total and direct effects of the NMR and smoking heaviness on lung cancer, CHD, COPD, and are shown in Fig 3 (ever smokers). These results are also included in S1 Table (lung cancer), S2 Table (CHD), and S3 Table (COPD) along with the F-statistics, Q-statistics and Egger intercept and the results among current, never and formers smokers. We have focussed on the

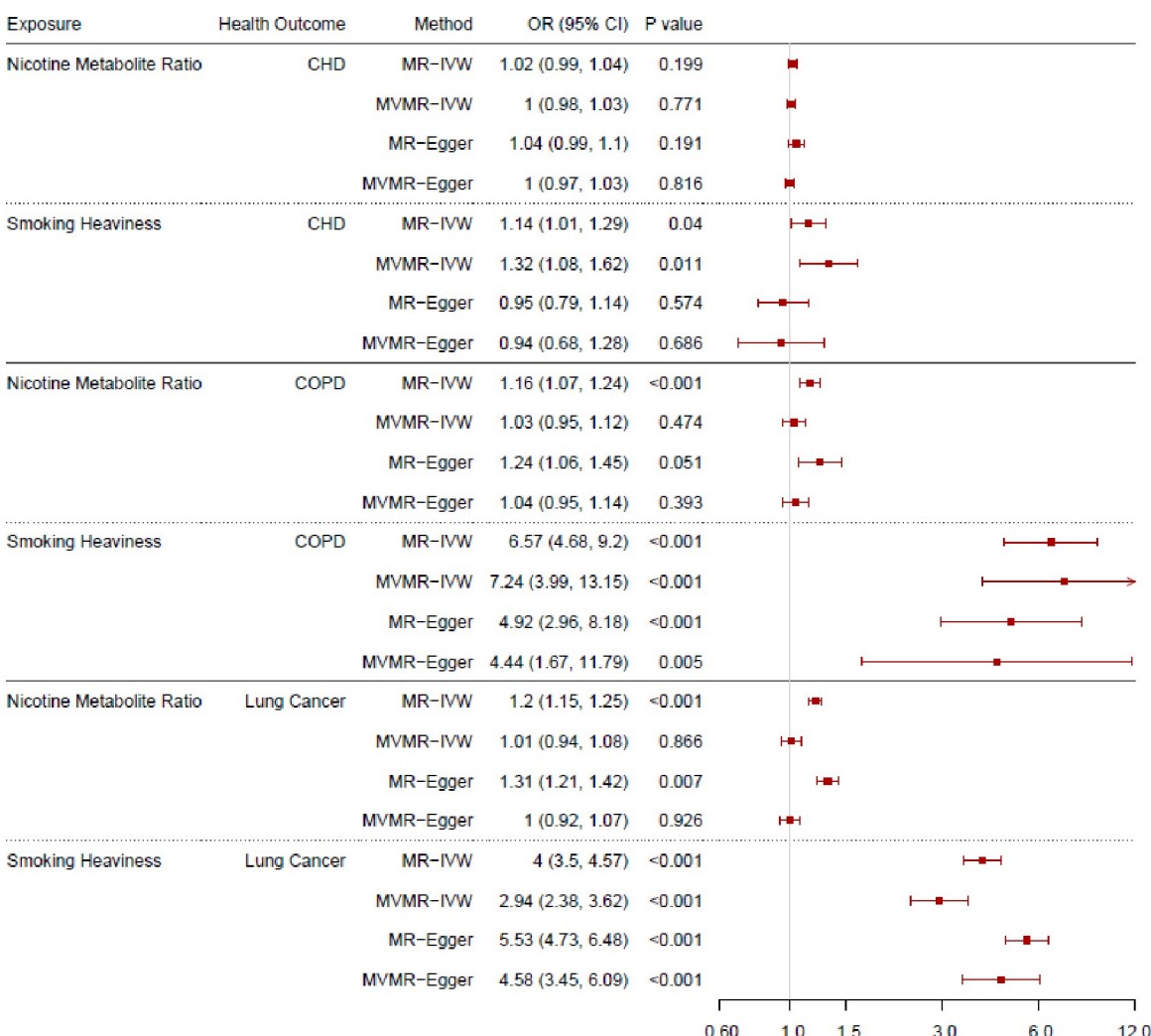

**Fig 3. Forest plot displaying the effect of nicotine metabolite ratio and smoking heaviness on coronary heart disease (CHD), chronic obstructive pulmonary disease (COPD) and lung cancer incidence: univariable Mendelian randomisation (MR) and multivariable Mendelian randomisation (MVMR) results among ever smokers.** Note: OR = Odds ratio.

results among ever smokers rather than current smokers here as receiving a diagnosis of lung cancer, CHD or COPD may increase the likelihood of someone quitting smoking, so the results are more relevant and interpretable among ever smokers. The results among current smokers can be found in S1 Note and S2 Fig. Additionally, we have included the median and mode weighted MR sensitivity analyses in S7 Table. Results are presented as odds ratios per standard deviation (SD) increase in the exposure phenotype (i.e., per SD increase in the NMR or cigarettes per day).

**Lung cancer.** The MR-IVW results indicate strong evidence to suggest that increased NMR and increased smoking heaviness both increase the risk of developing lung cancer among ever smokers (odds ratio [OR] = 1.20, 95% confidence interval [CI] 1.15 to 1.25; OR = 4.00, 95% CI 3.50 to 4.57 respectively). However, the MVMR-IVW results do not provide clear evidence to suggest an effect of NMR on lung cancer risk (OR = 1.01, 95% CI 0.94 to 1.08), and indicate that only increased smoking heaviness increased lung cancer risk (OR = 2.94, 95% CI 2.38 to 3.62). Although there is considerable evidence of heterogeneity and

potential pleiotropy in the smoking heaviness analyses among ever smokers, these results are supported by the MR-Egger results. Furthermore, there is no clear evidence of pleiotropy or bias due to population stratification indicated by the results among never smokers.

**Coronary heart disease.** The MR-IVW results provide no clear evidence that NMR affects CHD risk (OR = 1.02, 95% CI 0.99 to 1.04). This is supported by the MR-Egger results, but the weighted median and weighted mode results suggested there may be a weak effect (OR = 1.03, 95% CI 1.01 to 1.05, OR = 1.03, 95% CI 1.01 to 1.05 respectively). The results provide some evidence that increased smoking heaviness increases CHD risk (OR = 1.14, 95% CI 1.01 to 1.29) among ever smokers. The weighted median and weighted mode results support this finding (OR = 1.14, 95% CI 1.02 to 1.29, OR = 1.14, 95% CI 1.02 to 1.27 respectively); however, the MR-Egger analysis does not support this (OR = 0.95, 95% CI 0.79 to 1.14), and there is some evidence of heterogeneity indicated by the Q-statistics and the Egger intercept indicated weak directional pleiotropy which is supported by evidence of a protective effect among never smokers. The MVMR-IVW results provide no clear evidence of an effect of NMR on CHD risk (OR = 1.00, 95% CI 0.98 to 1.03) but some evidence that increased smoking heaviness increases CHD risk (OR = 1.32, 95% CI 1.08 to 1.62). As with the univariable results, the MVMR-Egger analyses do not support this (OR = 0.94, 95% CI 0.68 to 1.28) and there was evidence of heterogeneity and potential directional pleiotropy and some evidence of horizontal pleiotropy or bias due to population stratification in the analysis among never smokers.

**Chronic obstructive pulmonary disease.** The MR-IVW results indicate that increased NMR and smoking heaviness increase the risk of developing COPD among ever smokers (OR = 1.16, 95% CI 1.07 to 1.24, OR = 6.57, 95% CI 4.68 to 9.20 respectively). However, the MVMR-IVW results indicate no clear effect of NMR on COPD risk (OR = 1.03, 95% CI 0.95 to 1.12) but suggest that increased smoking heaviness increases COPD risk among ever smokers (OR = 7.24, 95% CI 3.99 to 13.15). These results are supported by the MR sensitivity analyses and MVMR-Egger results and there is no clear evidence of heterogeneity or directional pleiotropy or horizontal pleiotropy or bias due to population stratification among never smokers (where precise null effects were observed).

## Continuous outcomes

The total and direct effects of the NMR and smoking heaviness on lung function (FEV-1 and FVC) and heart rate are shown in Figs 4 (ever smokers) and 5 (current smokers). These results are also displayed in S4 Table (FEV-1), S5 Table (FVC), and S6 Table (heart rate) along with the F-statistics, Q-statistics and Egger intercept and the results among ever, current, never and formers smokers. Additionally, we have included the median and mode weighted MR sensitivity analyses in S7 Table Results are presented as betas per standard deviation (SD) increase in the exposure phenotype (i.e., per SD increase in the NMR or cigarettes per day). As per MR-STROBE guidelines [30], we have reported the results in text on an interpretable scale (i.e., difference in outcome in relevant units e.g., mL). However, to aid comparability across outcomes, we present the results per standard deviation change in the forest plots.

**Forced expiratory volume in 1 second.** The results relating to FEV-1 are presented in the text as changes in millilitres (mL) of expiration per standard deviation increase in the exposure. The MR-IVW results indicate that increased NMR and smoking heaviness both decrease FEV-1 among ever smokers (Fig 4, b = -13.35, 95% CI -18.38 to -8.33, b = -185.49, 95% CI -220.39 to -150.60 respectively). However, there is no clear evidence of an effect of NMR on FEV-1 in the MVMR-IVW analysis (b = 3.82, 95% CI -12.63 to 4.98), whereas there is evidence to suggest increased smoking heaviness decreased FEV-1 among ever smokers (b = -176.78, 95% CI -239.15 to -114.41). These results are supported by the MR sensitivity analyses and

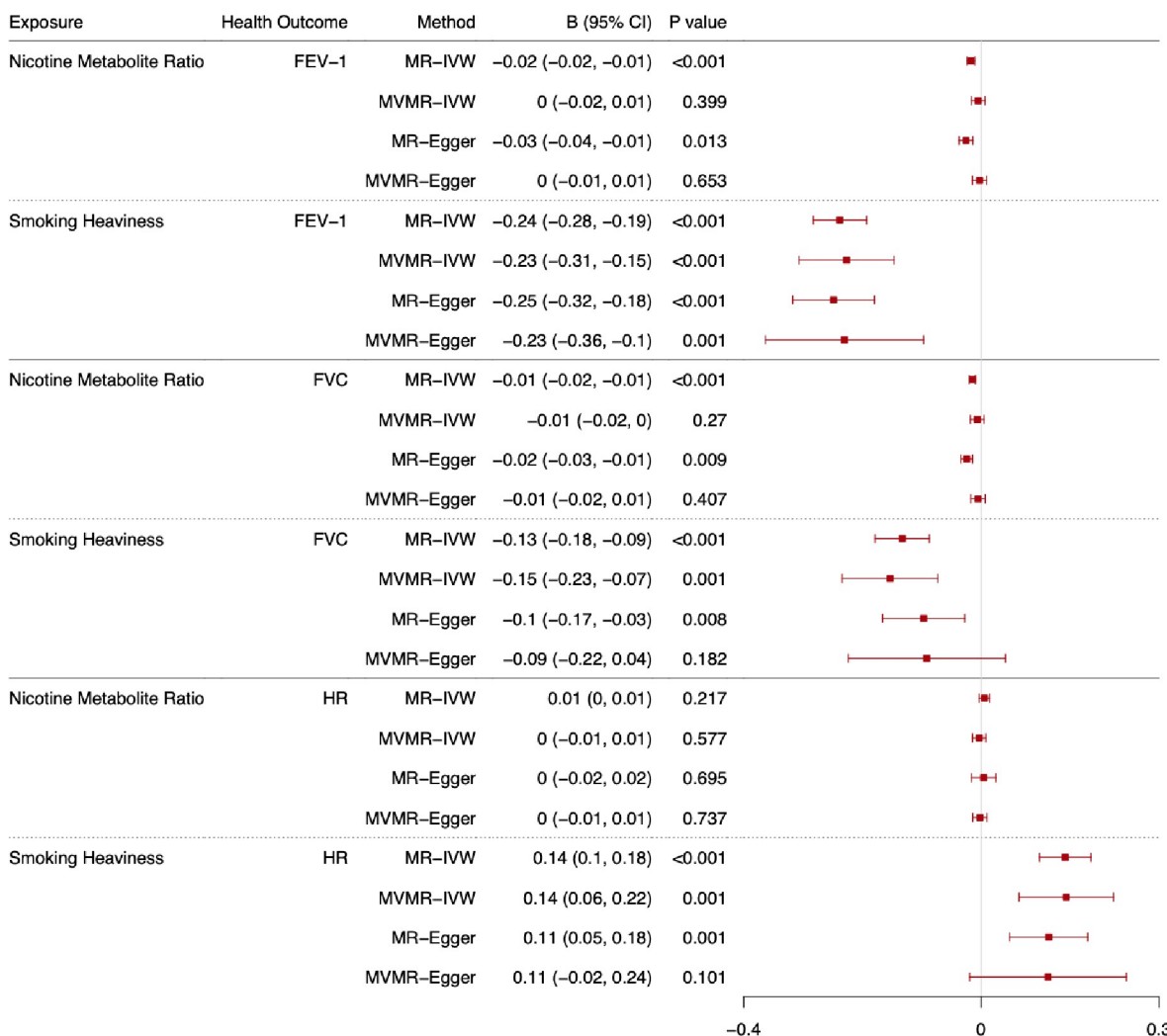

**Fig 4. Forest plot displaying the effect of nicotine metabolite ratio and smoking heaviness on standard deviations of lung function (FEV-1 and FVC), and heart rate (HR): univariable Mendelian randomisation (MR) and multivariable Mendelian randomisation (MVMR) results among ever smokers.** Note: B = beta.

MVMR-Egger results and there is no evidence of directional pleiotropy in the analyses, but there is evidence of heterogeneity in all analyses except for the NMR MR analyses. However, there is evidence of horizontal pleiotropy or bias due to population stratification in the NMR MR analyses among never smokers.

Among current smokers (Fig 5), the MR-IVW results indicate that increased NMR and smoking heaviness decreased FEV-1 (b = -33.33, 95% CI -41.76 to -24.90, b = -374.28, 95% CI -428.21 to -320.35 respectively). However, the MVMR-IVW results indicate a protective effect of nicotine exposure (i.e., lower nicotine exposure, due to higher NMR, lowers lung function) when smoking heaviness is accounted for among current smokers (NMR b = -17.77, 95% CI -29.92 to -5.62). The MVMR-IVW analyses support the MR-IVW results for smoking heaviness (b = -272.36, 95% CI -359.31 to -185.42). These results are supported by the MR sensitivity analyses and MVMR-Egger analyses. There is some evidence of heterogeneity in all analyses except the NMR MR analyses, directional pleiotropy in the MR analysis of smoking heaviness,

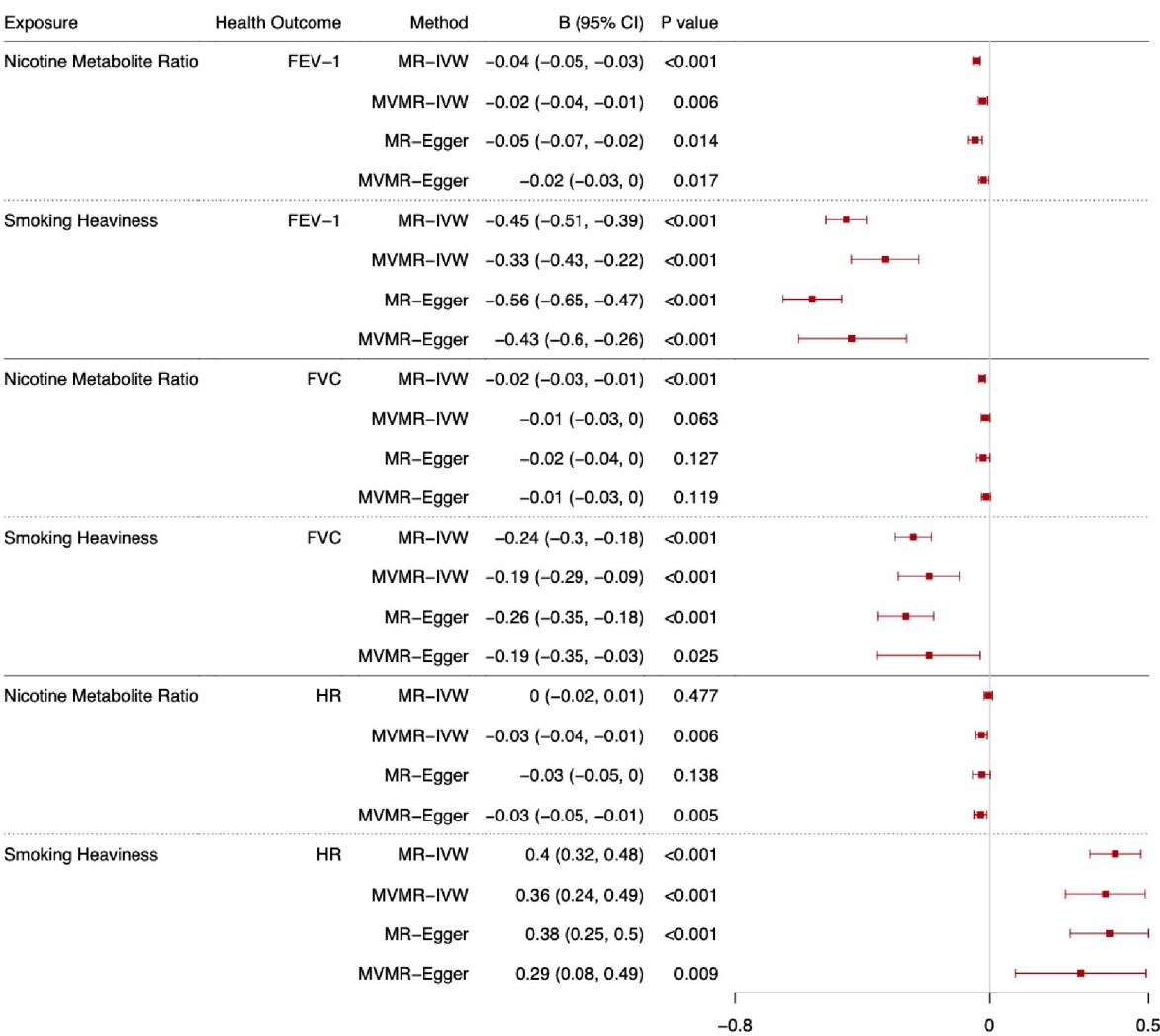

**Fig 5. Forest plot displaying the effect of nicotine metabolite ratio and smoking heaviness on standard deviations of lung function (FEV-1 and FVC), and heart rate (HR): univariable Mendelian randomisation (MR) and multivariable Mendelian randomisation (MVMR) results among current smokers.** Note: B = beta.

and horizontal pleiotropy or bias due to population stratification in the NMR MR analyses among never smokers.

**Forced vital capacity.** The results relating to FVC are presented in the text as changes in mL of capacity per standard deviation increase in the exposure. The MR-IVW results indicate that increased NMR and smoking heaviness decreased FVC among ever smokers (Fig 4, b = -14.34, 95% CI -19.42 to -9.26, b = -129.63, 95% CI -174.27 to -85.00 respectively). However, there is no clear evidence of an effect of NMR on FVC in the MVMR-IVW analysis (b = -6.32, 95% CI -17.41 to 4.77), whereas there is evidence to suggest increased smoking heaviness decreased FVC among ever smokers, after accounting for NMR (b = -149.55, 95% CI -228.07 to -71.03). These results are supported by the MR sensitivity analyses and MVMR-Egger results, although there is less clear evidence of an effect in the MVMR-Egger results for smoking heaviness than in the MR-Egger results. There is some evidence of heterogeneity in all analyses except the NMR MR analyses, but no clear evidence of directional pleiotropy. There is weak evidence to suggest potential horizontal pleiotropy / bias due to population

stratification in the NMR MR analyses and the smoking heaviness MR-IVW analysis among never smokers.

Among current smokers (Fig 5), the MR-IVW results suggest that increased NMR and smoking heaviness decreased FVC (b = -24.50, 95% CI -34.61 to -14.39, b = -246.13, 95% CI -304.36 to -187.89 respectively). MVMR-IVW results support the evidence for an effect of smoking cessation on FVC but indicate weak evidence of a protective effect of nicotine exposure when smoking heaviness is accounted for (NMR b = -13.50, 95% CI -27.39 to 0.39). The MR median and mode weighted analyses support these findings, and the MR-Egger and MVMR-Egger results are in the same direction but provide weaker support for the NMR results. There is evidence in the MVMR-IVW analysis to suggest that increased smoking heaviness decreases FVC (b -195.31, 95% CI -294.72 to -95.90), which is supported by the MVMR-Egger results. There is some evidence of heterogeneity in all analyses except the NMR MR analyses, but no clear evidence of directional pleiotropy. There is weak evidence to suggest potential horizontal pleiotropy or bias due to population stratification in the NMR MR analyses and the smoking heaviness MR-IVW analysis among never smokers.

**Heart rate.** The results relating to heart rate are presented in the text as changes in beats per minute per standard deviation increase in the exposure. Among ever smokers (Fig 4), the MR-IVW results suggest that increased NMR does not affect heart rate (b = 0.06, 95% CI -0.04 to 0.16), but increased smoking heaviness does increase heart rate (b = 1.61, 95% CI 1.12 to 2.10). The MVMR-IVW results support the results for both NMR and smoking heaviness (b = -0.04, 95% CI -0.16 to 0.09, b = 1.63, 95% CI 0.73 to 2.54 respectively). The MR sensitivity analyses and MVMR-Egger results also support these findings. There is some evidence of heterogeneity in all analyses, but no clear evidence of directional pleiotropy. There is weak evidence to suggest potential horizontal pleiotropy or bias due to population stratification in the MR-IVW NMR, MR-IVW and MR-Egger smoking heaviness analyses among never smokers.

Among current smokers (Fig 5), the MR-IVW results indicate no clear effect of NMR on heart rate (b = -0.05, 95% CI -0.20 to 0.09) but indicate that increased smoking heaviness increases heart rate (b = 4.58, 95% CI 3.65 to 5.51). However, the MVMR-IVW results suggest that decreased NMR when accounting for smoking heaviness (i.e., increased nicotine exposure per cigarette) increases heart rate among current smokers (b = -0.30, 95% CI -0.50 to -0.10) and increased smoking heaviness accounting for nicotine exposure increases heart rate (b = 4.22, 95% CI 2.77 to 5.68). The MR sensitivity and MVMR-Egger results support the MR-IVW and MVMR-IVW findings. There is some evidence of heterogeneity in all analyses except the MR NMR analyses, but no clear evidence of directional pleiotropy. There is weak evidence to suggest potential horizontal pleiotropy / bias due to population stratification in the MR-IVW NMR, MR-IVW and MR-Egger smoking heaviness analyses among never smokers.

## Interpreting the NMR MVMR results

In the MVMR results above, we interpret an increase in the direct effect of NMR as a reduction in nicotine exposure per cigarette smoked. S8 Table illustrates how the estimated direct effects of NMR found can be flipped to indicate the effect of increased nicotine exposure per cigarette smoked on the outcome.

## Sensitivity and supplementary analysis

The results of the sensitivity analyses whereby cotinine and cotinine plus 3HC were used as alternative proxies for nicotine exposure can be found in S1–S7 Tables The results are generally in line with the main results with some exceptions. Increased cotinine and cotinine plus 3HC exposure appears to increase risk of lung cancer (to a lesser extent than non-nicotine

constituents of tobacco smoke). There is also less evidence to suggest a protective effect of nicotine on COPD among current smokers in the analyses alternatively including cotinine and cotinine plus 3HC and there is less evidence of an effect on FEV-1 and heart rate among current smokers in the analyses including cotinine plus 3HC.

The results among former smokers can be found in S2–S7 Tables The findings suggest that there are likely lasting detrimental effects of smoking, but they are unlikely to be attributable to nicotine exposure.

## Discussion

Overall, we found little evidence to suggest a major detrimental effect of nicotine exposure on health when using an MVMR model to distinguish the direct effects of nicotine from the direct effects of non-nicotine components of tobacco smoke (using NMR and CPD as proxies). Our model suggests that, among current smokers, increased nicotine exposure per cigarette smoked increases heart rate but may have protective effects on lung function. Our findings in relation to smoking heaviness were generally as expected except for CHD, for which there was limited evidence of an effect. In line with previous research [20–22], increased smoking heaviness detrimentally impacted risk of lung cancer, COPD, and CHD, as well as lung function and heart rate. As expected, and in line with evidence from a range of human and animal studies [26,28], our analysis suggests that exposure to nicotine without the remaining constituents of tobacco smoke increases heart rate (the positive control).

In contrast to previous evidence, in which increased smoking heaviness was found to increase risk of CHD, we found limited evidence of an effect of smoking heaviness on CHD among ever smokers. Previous MR studies have also shown minimal evidence of this effect using UK Biobank data [24], which could indicate that the outcome GWAS sample may not be representative of the general population (i.e., selection or collider bias may have impacted our results). Indeed, UK Biobank participants are more likely to describe themselves as female, have fewer health issues, live in less socioeconomically deprived areas, and be more highly educated, than the general population and thus the results may suffer from "healthy volunteer" selection bias [31]. Given that CHD may disproportionately affect men [32] and people with generally poorer health, this may explain the limited findings. Additionally, we may have had limited power to detect this effect given the range of causes of CHD and relatively few cases in UK Biobank.

Another unexpected finding was the apparent protective effects of nicotine on lung health (COPD, FEV-1 and FVC) among current smokers. However, previous evidence suggests those with a lower NMR (and greater circulating nicotine per cigarette) inhale less smoke and other non-nicotine constituents per cigarette, and have lower levels of biomarkers of inflammation [18,19,28]. Our MVMR model is designed to estimate nicotine exposure, but measuring smoking heaviness by the number of cigarettes smoked per day will not completely capture the variation in smoking intensity [33], which could complicate the interpretation of our findings. There could be a small effect of intensity which is captured in the instrumental variable for NMR because increased NMR may lead to increased inhalation per cigarette. Given the magnitude of effects found for NMR are much smaller than for smoking heaviness, the residual variance in smoking intensity impacting the NMR estimates appears to be limited. Furthermore, if there is a true protective effect it may not be clinically important given the magnitude of the effects found. For example, patients only perceive differences in FEV-1 with changes of 112 mL [34] whereas the MVMR-IVW analyses indicated an increase of only 17.77 mL per SD increase in NMR.

In the supplementary analyses, we also saw apparent effects of cotinine and cotinine plus 3HC on lung cancer risk in the MVMR models. Given that nicotine is not thought to be

carcinogenic [27], these findings could indicate pleiotropic pathways are involved for these additional exposures (e.g., via metabolism). This highlights that NMR may be a more appropriate proxy for nicotine exposure in this context; NMR is a measure of nicotine metabolism, therefore by including it in an MVMR model which is robust to pleiotropy via each exposure in the model, we account for pleiotropic pathways via metabolism.

Although this is the first study to use MVMR to explore the impact of increased daily nicotine inhalation in humans, the study is not without limitations. The study was designed to explore the impact of long-term nicotine use, but the exposures included may not reflect long-term use. However, the current smokers included in our UK Biobank GWAS had smoked a mean of 28 pack years (median = 25 pack years, interquartile range = 15 to 37 pack years), and less than 5% of the sample had less than 4 pack years of exposure. The study is limited by the potential presence of horizontal pleiotropy in some analyses (as indicated by effects seen in the analyses restricted to non-smokers and the Egger analyses), which violates an assumption of the method. The effect estimates among never smokers cannot be meaningfully interpreted as we know that never smokers (or at least those who accurately self-report never smoking) do not smoke any cigarettes per day despite being predisposed to heavier smoking, but evidence of an effect in never smokers (along with a high Cochran's Q statistic) is indicative of horizontal pleiotropy. Despite this, no effects were observed in the relationship between NMR and the health outcomes in the MVMR analyses restricted to never smokers. However, stratifying data in this manner can lead to collider bias [35]. Additionally, while the MR-Egger test of directional pleiotropy indicated directional pleiotropic effects in the relationship between smoking heaviness and health outcomes, there was no clear evidence of directional horizontal pleiotropic effects in the relationship between NMR and health outcomes among ever and current smokers. Therefore, these sensitivity analyses suggest our interpretation of the direct effects of nicotine on the selected health outcomes should not be impacted by directional horizontal pleiotropy or population stratification.

Further limitations of the study relate to the adjustment and inclusion criteria of the GWAS used for analysis. The first issue pertains to differences between the exposure GWAS inclusion criteria: the GWAS of NMR was restricted to current smokers whereas the GWAS of smoking heaviness was restricted to ever regular smokers. However, smoking heaviness was assessed as current or past smoking heaviness (i.e., the measure is representative of the behaviour of current smokers despite being a retrospective measure, therefore we are assuming that the two populations are similar). If this assumption is incorrect, we may have introduced bias into our findings. Second, in contrast to the GWAS of smoking behaviour and the health outcomes, the NMR GWAS was adjusted for body mass index (BMI). Adjusting for BMI in one exposure variable meant that we could not clearly assess the effect of nicotine on BMI, however it also may impact our interpretation of results where the outcomes are associated with BMI (i.e., where BMI is a plausible co-variate of the exposure and outcome phenotypes) [36]. As lung and heart health could be impacted by BMI [37,38], some of the effect sizes observed may be biased towards the null. Although there are approaches available to correct for this bias [39], it is not feasible to implement this correction with the limited number of SNPs included in this analysis. Therefore, we must interpret these findings with caution. Third, we did not use data from an updated CPD GWAS with a larger sample size and more diverse sample that was released in 2022 [40] after analysis was complete for this study. The study team agreed that it would be detrimental to use the new data release restricted to European ancestry. The number of SNPs found to be associated with smoking heaviness was three times greater in the 2022 data release compared to the 2019 data release, which would cause a problematic imbalance/mismatch between the number of SNPs used to instrument smoking heaviness versus NMR, and potentially lead to weak instrument bias [41]. Furthermore, we were unable to use data from the

multi-ancestry GWAS of smoking heaviness [40] as the NMR GWAS did not conduct similar analyses and was restricted to those of European descent and an assumption of MVMR is that all data included in an MVMR model are from the same underlying population [13]. Future work could focus on the inclusion of multiple ancestries if appropriate NMR GWAS become available to explore whether the findings generalise to non-European ancestries.

This work highlights the limited impact that increased daily nicotine exposure via inhalation via smoking is likely to have on heart and lung health. However, further research is needed to understand the health impact of increased daily exposure to nicotine via other sources (e.g., e-cigarettes) which contain other components.

## Conclusions

In conclusion, the present findings indicate that increased daily nicotine exposure (via inhalation) when accounting for exposure to non-nicotine constituents of tobacco smoke may increase heart rate but does not increase risk of COPD or CHD and does not appear to adversely impact lung function. We found that, aside from effects on heart rate (which were expected given our knowledge of effects of short-term nicotine use), there was no evidence to suggest that nicotine exposure is responsible for the detrimental effects of smoking on the outcomes that were included in this analysis. Although further research is necessary to explore other health outcomes and triangulate these findings, our results support existing evidence which suggests nicotine use is not a major risk factor in the development of smoking-related disease.

## Methods

### Ethics statement

UK Biobank received ethics approval from the Research Ethics Committee (REC reference for UK Biobank is 11/NW/0382). Written informed consent was obtained from participants prior to their participation in UK Biobank. Approval to use these data was sought from and approved by UK Biobank (project 9142). All studies approved by UK Biobank do not require individual ethical clearance as UK Biobank has approval from the North West Multi-centre Research Ethics Committee as a Research Tissue Bank under which each approved study operates.

### Data sources

To identify relevant instrumental variables to include in the model, we sourced summary-level genetic data from well-powered, published GWAS. We identified SNPs associated with NMR, cotinine, cotinine plus 3HC and smoking heaviness (measured by CPD) [14,15,42]. For the outcomes, we required data to be stratified by smoking status in order to be comparable with the population of the exposure GWAS data. An appropriately stratified GWAS (stratified by smoking status: ever versus never [43]) was only available for one of the outcomes, lung cancer. For the six remaining outcomes, we were not aware of any existing available GWAS summary-level data that appropriately stratified by smoking status. We therefore used UK Biobank data to conduct GWAS of CHD, COPD, FEV, FVC and HR, stratified by smoking status and restricted to individuals of European ancestry. These analyses were: 1) stratified by whether participants had ever smoked, and further stratified into 2) current smoker, or 3) former smoker, with 4) never smoker as the comparator, and were conducted using the MRC Integrative Epidemiology Unit UK Biobank GWAS pipeline (version 2) [44,45]. We also conducted stratified GWAS of BMI using UK Biobank data with the intention of including BMI as an outcome. However, these methods and results are not presented here as we identified a potential

interpretation issue: the NMR GWAS was adjusted for BMI, meaning these results would likely be biased [36].

**Smoking heaviness.** Liu and colleagues [14] report summary-level statistics from a GWAS meta-analysis of smoking heaviness (measured by standard deviation change in CPD categories, equivalent to 2–3 additional cigarettes per day) among 337,334 smokers of European ancestry in the GWAS and Sequencing Consortium of Alcohol and Nicotine (GSCAN). Smokers were defined as having ever or currently smoked and smoking heaviness (at the time of smoking) was binned into five categories or included as pre-defined bins from the original study (see supplementary material in Liu and colleagues [14]). Analyses were adjusted for age, age-squared, sex, genetic principal components, and smoking status (current versus former). SNPs associated with CPD explained 4% of variance in CPD. SNPs were reported as independent if they explain additional variance in conditional analyses using a partial correlation-based score statistic [46]. The reported data are available at: https://doi.org/10.13020/3b1n-ff32. To avoid sample overlap, two versions of the GSCAN data were used in our analyses. Where we analysed UK Biobank outcomes, we used a version of the GSCAN data with UK Biobank removed to avoid sample overlap. To maximise sample size, we used GSCAN data including UK Biobank in analyses which did not include UK Biobank in the outcome GWAS. In all analyses, 23andMe were removed from the GSCAN data due to data sharing restrictions.

**Nicotine.** Buchwald and colleagues [15] report summary-level statistics from a GWAS meta-analysis of the standard deviation change in NMR among 5,185 current smokers of European descent with cotinine levels ≥10 ng/ml (indicating recent smoking). NMR is a ratio of 3'hydroxycotinine/cotinine which indicates how quickly a person metabolises and clears nicotine. Analyses were adjusted for population substructure, age, sex, BMI, alcohol use, and birth year. The SNPs associated with NMR explained ~38% of the variance in NMR. SNPs were reported as independent if they explained additional variance in a step-wise conditional regression using genome-wide complex trait analysis [47]. Information relating to the nicotine measures used in the supplemental analysis (using cotinine and cotinine plus 3HC data as a proxy for nicotine exposure) can be found S2 Note.

**Lung cancer.** McKay and colleagues [43] report summary-level statistics from a case-control GWAS meta-analysis of overall lung cancer risk (including adenocarcinoma, squamous cell carcinoma, and small cell carcinoma) among 50,046 individuals of European descent using the International Lung Cancer Consortium (ILCCO). Analyses were adjusted for age, gender, country (if applicable) and significant principal components. We received the data via request to the authors and we additionally received the data stratified by smoking status (ever versus never smoking). Definitions of ever and never smoking differed by cohort included in the meta-GWAS. Details of the definitions can be found in McKay and colleagues' [43] supplementary note.

**Non-cancer smoking-related health outcomes.** We conducted GWAS of non-cancer health outcomes using data from UK Biobank, a population-based health research resource consisting of approximately 500,000 people, aged between 38 years and 73 years, who were recruited between the years 2006 and 2010 from across the UK [48]. UK Biobank has a particular focus on identifying determinants of human diseases in middle-aged and older individuals who provided a wide range of health information (data available at www.ukbiobank.ac.uk). A full description of the study design, participants, and quality control (QC) methods have been described previously [48,49].

The full data release contains the cohort of successfully genotyped samples (n = 488,377). Further information about genotyping and imputation can be found in S3 Note. Individuals with mismatched sex were excluded and the analyses were restricted to those of 'European' ancestry using an in-house k-cluster means method. The GWAS were conducted using the

linear mixed model (LMM) association method as implemented in BOLT-LMM (v2.3) [50], adjusting for genotype array, age and sex [44]. BOLT-LMM association statistics are on the linear scale, so test statistics (betas and their corresponding standard errors) relating to binary phenotypes were transformed to log odds ratios and their corresponding 95% confidence intervals (95% CI) on the liability scale using a Taylor transformation expansion series in which we divided the statistics by $\mu(1 - \mu)$, whereby $\mu$ is the case prevalence [51]. Each outcome was stratified by self-reported participant smoking status (ever [further stratified by current or former] and never), resulting in four GWAS per outcome. Smoking status was categorised as never, previous, and current smoking in UK Biobank (field ID 20116). From this variable, we derived an 'ever smokers' category which was defined as currently or having previously smoked occasionally, most days or daily (i.e., having smoked more than just once or twice). Current smoking was defined as currently smoking occasionally, most days or daily. Former smoking was defined as not currently smoking but having previously smoked occasionally, most days or daily (i.e., more than just once or twice). Those who had tried smoking once or twice or who had never smoked were categorised as never smokers.

We identified COPD cases as participants who self-reported a doctor's diagnosis of COPD. Lung function (FEV-1 and FVC, in litres) was measured using a Vitalograph spirometer. CHD diagnosis was determined using linked hospital admission data (ICD codes relating to Ischemic Heart Disease). Further information regarding each health outcome (including UK Biobank field IDs) can be found in the S4 Note.

## Statistical analysis

All analyses were conducted using R version 4.1.1. The statistical analysis was completed using the TwoSampleMR, MVMR and MendelianRandomization packages, and the statistical code and data used is available online (https://doi.org/10.5281/zenodo.10469666).

**Selection of genetic variants.** We selected conditionally independent (at the genome-wide significant level, $p < 5 \times 10^{-8}$) genetic variants identified in the conditional GWAS analyses of NMR and CPD for inclusion in the analysis [14,15]; 55 SNPs associated with smoking heaviness were identified as conditionally independent [14]; 7 SNPs associated with the NMR were identified as conditionally independent. Methods of determining conditional independence varied between the two exposure GWAS, whereby GSCAN reported SNPs as independent if they explain additional variance in conditional analyses using a partial correlation-based score statistic [14] and Buchwald and colleagues reported SNPs as conditionally independent if they explained additional variance in a step-wise conditional regression using genome-wide complex trait analysis [15].

After removing SNPs which were not available in the outcome GWAS (where no available proxy could be identified), harmonising, and clumping the combined exposure datasets with the outcome dataset, the number of SNPs in each analysis varied. S9–S14 Tables detail which SNPs were included and excluded (with reasons) and the total number of SNPs in each analysis. S5 Note gives further details of the proxy search and clumping methods.

We tested instrument strength and validity using the two-sample conditional F-statistic for MVMR and Cochran $Q$ statistic [13,41]. The conditional F-statistic for MVMR indicates instrument strength of each exposure when accounting for the prediction of other exposures in the model (i.e., whether the SNPs jointly predict smoking heaviness after predicting the NMR, and vice versa) [41]. The standard critical values for the F-statistic and Q-statistic can be approximated so F-statistic should be greater than 10 to indicate sufficient instrument strength and $Q$ estimates should be less than the number of SNPs included in the model to indicate no excessive heterogeneity [52,53].

**Univariable Mendelian randomisation.** For comparison with the MVMR analysis, we considered the total effect of both the NMR and smoking heaviness on each health outcome using univariable MR. We used four complementary MR methods MR-IVW, MR-Egger, weighted median, weighted mode). Details of the univariable MR analysis methods can be found in S6 Note.

**Multivariable Mendelian randomisation.** We explored the direct effects of both the NMR and smoking heaviness on health outcomes using MVMR. We used summary data from Buchwald and colleagues' [15], GSCAN [14] and ILCCO [43], and UK Biobank GWAS pipeline results. We repeated these analyses using two complementary methods–MVMR-IVW and MVMR-Egger [13,29].

Given that the GWAS of smoking heaviness was restricted to ever smokers and the GWAS of the NMR was restricted to current smokers, the analyses were restricted to ever and current smokers as the summary statistics are not applicable to never smokers. However, the ILCCO lung cancer data have only been stratified by ever and never smoking status; therefore, analyses restricted to current smokers were not possible when exploring lung cancer incidence. In supplementary analyses, we additionally stratified the analysis by never smokers to explore potential horizontal pleiotropy–effects observed among never smokers could indicate horizontally pleiotropic effects (i.e., the included SNPs influencing the outcome directly, or via another phenotype, but not through the measured exposure), misreporting of smoking status, or residual population stratification. Horizontally pleiotropic genetic variants are not valid instruments in MR analyses and their inclusion would result in a violation of the exclusion restriction assumption (further details on the assumptions of MR can be found in S7 Note). Pleiotropy robust methods (e.g., MR-Egger) are robust to horizontal pleiotropy under assumptions of the form that pleiotropy takes. For all outcomes except lung cancer (where data are unavailable), we also included supplementary results restricted to former smokers to explore whether any health effects found among current smokers may be recoverable (i.e., not present among former smokers who have recovered following smoking cessation).

## Supporting information

**S1 Note. S1 Note details the results of the analyses exploring the effect of nicotine and non-nicotine constituents of tobacco smoke on coronary heart disease and chronic obstructive pulmonary disease among current smokers.**
(DOCX)

**S2 Note. S2 Note describes further information regarding the nicotine measures used in the main and supplementary analysis.**
(DOCX)

**S3 Note. S3 Note provides further information about the genotyping, quality control, and imputation methods for the UK Biobank data.**
(DOCX)

**S4 Note. S4 Note provides further information regarding the health outcomes used in the main and supplementary analysis (including UK Biobank outcome field codes).**
(DOCX)

**S5 Note. S5 Note provides further details on how proxy SNPs were identified and the clumping methodology.**
(DOCX)

**S6 Note.** S6 Note provides details of the univariable Mendelian randomisation methods.
(DOCX)

**S7 Note.** S7 Note details the assumption of Mendelian randomisation.
(DOCX)

**S1 Table.** Univariable and multivariable Mendelian randomisation IVW and Egger analysis of nicotine exposure proxies (nicotine metabolite ratio, cotinine and cotinine+-3'hydroxycotinine) and smoking heaviness (cigarettes per day) and lung cancer.
(XLSX)

**S2 Table.** Univariable and multivariable Mendelian randomisation IVW and Egger analysis of nicotine exposure proxies (nicotine metabolite ratio, cotinine and cotinine+-3'hydroxycotinine) and smoking heaviness (cigarettes per day) and coronary heart disease.
(XLSX)

**S3 Table.** Univariable and multivariable Mendelian randomisation IVW and Egger analysis of nicotine exposure proxies (nicotine metabolite ratio, cotinine and cotinine+-3'hydroxycotinine) and smoking heaviness (cigarettes per day) and chronic obstructive pulmonary disease.
(XLSX)

**S4 Table.** Univariable and multivariable Mendelian randomisation IVW and Egger analysis of nicotine exposure proxies (nicotine metabolite ratio, cotinine and cotinine+-3'hydroxycotinine) and smoking heaviness (cigarettes per day) and forced expiratory volume in 1 second (litres).
(XLSX)

**S5 Table.** Univariable and multivariable Mendelian randomisation IVW and Egger analysis of nicotine exposure proxies (nicotine metabolite ratio, cotinine and cotinine+-3'hydroxycotinine) and smoking heaviness (cigarettes per day) and forced vital capacity (litres).
(XLSX)

**S6 Table.** Univariable and multivariable Mendelian randomisation IVW and Egger analysis of nicotine exposure proxies (nicotine metabolite ratio, cotinine and cotinine+-3'hydroxycotinine) and smoking heaviness (cigarettes per day) and heart rate (beats per minute).
(XLSX)

**S7 Table.** Univariable Mendelian randomisation weighted mean and weighted Mode analysis of nicotine exposure proxies (nicotine metabolite ratio, cotinine and cotinine+-3'hydroxycotinine) and smoking heaviness (cigarettes per day) on smoking related health outcomes.
(XLSX)

**S8 Table.** A table to aid interpretation of the multivariable Mendelian randomisation analysis of the direct effect of nicotine metabolite ratio (accounting for smoking heaviness).
(XLSX)

**S9 Table.** Details of the inclusion and exclusion of Single Nucleotide Polymorphisms (SNPs) in the analysis exploring the effect of nicotine metabolite ratio and smoking heaviness on lung cancer.
(XLSX)

**S10 Table. Details of the inclusion and exclusion of Single Nucleotide Polymorphisms (SNPs) in the analysis exploring the effect of cotinine and smoking heaviness on lung cancer.**
(XLSX)

**S11 Table. Details of the inclusion and exclusion of Single Nucleotide Polymorphisms (SNPs) in the analysis exploring the effect of cotinine plus 3'hydroxycotinine (3HC) and smoking heaviness on lung cancer.**
(XLSX)

**S12 Table. Details of the inclusion and exclusion of Single Nucleotide Polymorphisms (SNPs) in the analysis exploring the effect of nicotine metabolite ratio and smoking heaviness on UK Biobank smoking-related health outcomes.**
(XLSX)

**S13 Table. Details of the inclusion and exclusion of Single Nucleotide Polymorphisms (SNPs) in the analysis exploring the effect of cotinine and smoking heaviness on UK Biobank smoking-related health outcomes.**
(XLSX)

**S14 Table. Details of the inclusion and exclusion of Single Nucleotide Polymorphisms (SNPs) in the analysis exploring the effect of cotinine plus 3'hydroxycotinine (3HC) and smoking heaviness on UK Biobank smoking-related health outcomes.**
(XLSX)

**S1 Fig. A diagram to show the by-products of nicotine metabolism. Adapted from Benowitz, Hukkanen, & Jacob, 2009 [16].**
(TIF)

**S2 Fig. Forest plot displaying the effect of nicotine metabolite ratio and smoking heaviness on coronary heart disease (CHD) and chronic obstructive pulmonary disease (COPD): univariable Mendelian randomisation (MR) and multivariable Mendelian randomisation (MVMR) results among current smokers.** Note: OR = Odds ratio.
(TIF)

## Acknowledgments

We would like to thank Rachel Tyndale and her team for their support in interpreting the MVMR results relating to NMR and provision of summary statistics. We would also like to thank the GSCAN and ILCCO teams (including James McKay) for the provision of summary statistics. Quality Control filtering of the UK Biobank data was conducted by R. Mitchell, G. Hemani, T. Dudding, L. Corbin, S. Harrison, L. Paternoster as described in the published protocol (doi: 10.5523/bris.1ovaau5sxunp2cv8rcy88688v). The MRC IEU UK Biobank GWAS pipeline was developed by B. Elsworth, R. Mitchell, C. Raistrick, L. Paternoster, G. Hemani, T. Gaunt (doi: 10.5523/bris.pnoat8cxo0u52p6ynfaekeigi). This research has been conducted using the UK Biobank Resource under Application Number 9142.

## Author Contributions

**Conceptualization:** Jasmine N. Khouja, Eleanor Sanderson, Amy E. Taylor, Marcus R. Munafò.

**Data curation:** Jasmine N. Khouja.

**Formal analysis:** Jasmine N. Khouja.

**Funding acquisition:** Jasmine N. Khouja, Rebecca C. Richmond, Marcus R. Munafò.

**Investigation:** Jasmine N. Khouja, Billy A. Church.

**Methodology:** Jasmine N. Khouja, Eleanor Sanderson, Robyn E. Wootton, Amy E. Taylor, Rebecca C. Richmond, Marcus R. Munafò.

**Supervision:** Eleanor Sanderson, Robyn E. Wootton, Amy E. Taylor, Rebecca C. Richmond, Marcus R. Munafò.

**Visualization:** Jasmine N. Khouja, Billy A. Church.

**Writing – original draft:** Jasmine N. Khouja.

**Writing – review & editing:** Jasmine N. Khouja, Eleanor Sanderson, Robyn E. Wootton, Amy E. Taylor, Billy A. Church, Rebecca C. Richmond, Marcus R. Munafò.

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
