## [Decision Letter · Decision Letter 0]

3 Oct 2023

Dear Dr Khouja,

Thank you very much for submitting your Research Article entitled 'Estimating the long-term health impact of nicotine exposure by dissecting the effects of nicotine versus non-nicotine constituents of tobacco smoke: A multivariable Mendelian randomisation study' to PLOS Genetics.

The manuscript was fully evaluated at the editorial level and by independent peer reviewers. The reviewers appreciated the attention to an important problem, but raised some substantial concerns about the current manuscript. Based on the reviews, we will not be able to accept this version of the manuscript, but we would be willing to review a much-revised version. We cannot, of course, promise publication at that time.

If you decide to revise the manuscript for further consideration at PLOS Genetics, please aim to resubmit within the next 60 days, unless it will take extra time to address the concerns of the reviewers, in which case we would appreciate an expected resubmission date by email to plosgenetics@plos.org.

We are sorry that we cannot be more positive about your manuscript at this stage. Please do not hesitate to contact us if you have any concerns or questions.

Yours sincerely,

Zoltán Kutalik, PhD

Academic Editor

PLOS Genetics

Scott Williams

Section Editor

PLOS Genetics

First, I'd like to apologies for the lengthy review process. As the authors can see from the reviewers' comments the manuscript represents interesting and innovative research with potential to push our knowledge further in separating the impact of nicotine vs non-nicotine-based smoking exposures in disease. The reviews are extensive, but addressable and observational (longitudinal) reanalysis would greatly improve the paper.

Reviewer's Responses to Questions

**Comments to the Authors:**

Reviewer #1: This manuscript analyses the effects of smoking on lung diseases using a Mendelian Randomization approach. The authors use genetic proxies related to self-reported cigarettes consumed per day along with genetic proxies for nicotine exposure/metabolism to obtain estimates of direct effects of smoking and indirect effects of nicotine exposures on lung disease risks. The CPD trait has very low heritability in general while the nicotine metabolism trait has high heritability. Ordinarily I think the approach that has been taken would be rather unreliable due to low heritability of CPD but because the factor being conditioned on has much higher heritability the general framework seems likely to give reasonable estimates (but there have never been simulation studies of this kind of analysis to indicate reliability). The quality statistics for MR and multivariable MR appear acceptable. The analysis appears to be reasonable and the results are interesting.

While the analyses and results are well presented, the interpretation is completely irrelevant to the data that have been analyzed. There is no information about e-cigarettes and we have no direct information about smoking over time. The genetic instruments are proxies for behavior that may or may not happen and the low heritability of cigarettes per day mean that actual behaviors can have substantial impacts over the lifecourse that cannot be modeled in the way they have approached the analysis. That said, if the authors have access to the original UK Biobank data, since it is now a longitudinal study, and direct measures of smoking intensity along with the diseases they are studying are available, it would be possible to further analyze UKBB data to evaluate if the observations they are inferring from genetic proxies are reflected in the direct analyses that are possible in UKBB. Using UKBB as a primary source for these studies might not be optimal given limited numbers of lung cancers in this resource, but for validating findings such an analysis would be useful. On the otherhand if the authors do not want to directly analyze smoking, nicotine and outcomes using direct multivariate studies, I think the paper has sufficient merit based on the analyses that have been completed to warrant consideration as a publication. However, the interpretation needs to be very carefully and almost completely rewritten.

For example, the discussion from lines 538-551 is not supported by any of the data that have been analyzed, because the CHD proxies are constructed at birth based on genetic profiles and their is no information in this kind of analysis as it has been conducted to evaluate risk for CHD over the lifespan. One might theoretically construct genetic proxies for early and late age CHD and then that might give some inference about lifespan effects, but the authors have not done that so cannot make any speculative comments about lifecourse events from the analyses they have done.

Similarly, there is absolutely no information in this study about e-cigarettes. Therefore, there is no relevance of any of the discussion about them. Aside from nicotine in e-cigarettes, there are many other compounds that do influence lung diseases. Therefore, all of the comments about risks from e-cigarettes are speculative and not tied to the analyses that were conducted. The All of Us cohort does collect data on e-cigarette use, so the authors could evaluate them from that cohort if they would choose to do that. Otherwise, they should remove all of the discussion of e-cigarette use in most of the paper. It may be possible to retain one sentence restricting commentary about e-cigarettes to just the impact that the nictoine component would have but the rest of the commentary needs to be removed to ensure consistency of the findings and their discussion with what was actually done in the analysis.

Starting with line 610 we again get an interpretation untied to any of the analyses that were done. The authors have modeled CPD and nicotine metabolism but then discuss long-term nicotine exposure via inhalation. If the authors had analyzed smoking duration or packyears of exposure with genetic proxies perhaps this paragraph would make some sense, but they analyzed CPD which is a one time measure unrelated to lifecourse. The entire paragraph should be removed.

Similarly the discussion which describes many factors that were not studied needs to be completely rewritten.

The kind of excessive extrapolations that are made in this manuscript can cause more harm than good in the literature if they are not significantly adjusted. While the ultimate message may be correct, the process taken to get to the message has to be accurately supported by analyses.

Reviewer #2: The authors conducted a multivariable Mendelian randomization study to explore the direct effects of nicotine compared with the non-nicotine constituents of tobacco smoke on health outcomes. I find the study quite interesting with, however, several points that need to be refined.

1) Abstract needs to be rewritten to increase clarity. For instance, more details are needed around GWAS used for the MV-MR analyses (data source, number of variants per trait, etc.). Also, line 28 starts with “these results suggest” although no results have been presented.

2) Line 122: the exposure used in the MV-MR needs to be added.

3) Lines 123-125: further rational on the outcomes explored needs to be provided.

4) Line 136: A clarification on selection criteria for the variants used for each analysis needs to be added.

5) Line 143: A further discussion on sample overlap needs to be added. Exposure and outcome data are estimated in the UK Biobank and results obtained may be biased.

6) Line 165: Betas and SEs for all analyses performed need to be added in supplementary material. It is not clear from the description of the methods what is number of variants used for each analysis.

7) Line 215: Further clarification on the conversion of results obtained from BOLT to reflect Odds Ratios needs to be added. For instance, was prevalence of the outcome accounted for in the formula used?

8) Line 213: Package used for the analyses need to be added as well as codes posted on a repository would be of help for the findings to be reproducible.

9) Line 237: Please clarify what is meant by “conditional”.

10) Line 243: How many SNPs where removed from each analysis?

11) Line 260: Please add univariable MR to increase clarity.

12) Line 306: Cochran’s Q statistics and p-values need to be added to show if heterogeneity is significant across various effect estimates.

13) Figure 3. A further explanation on why you observe a stronger association of smoking heaviness-COPD in the MV analyses compared to univariable analyses is needed.

14) Figure 5. Please clarify “B” in the figure legend.

**Have all data underlying the figures and results presented in the manuscript been provided?**

Reviewer #1: Yes

Reviewer #2: None

PLOS authors have the option to publish the peer review history of their article (what does this mean?). If published, this will include your full peer review and any attached files.

Reviewer #1: **Yes: **Christopher Amos

Reviewer #2: No

---

## [Decision Letter · Decision Letter 1]

29 Jan 2024

Dear Dr Khouja,

We are pleased to inform you that your manuscript entitled "Estimating the health impact of nicotine exposure by dissecting the effects of nicotine versus non-nicotine constituents of tobacco smoke: A multivariable Mendelian randomisation study" has been editorially accepted for publication in PLOS Genetics. Congratulations!

Yours sincerely,

Zoltán Kutalik, PhD

Academic Editor

PLOS Genetics

Scott Williams

Section Editor

PLOS Genetics

Comments from the reviewers (if applicable):

Reviewer's Responses to Questions

**Comments to the Authors:**

Reviewer #1: The authors have completely revised this manuscript and addressing all of my concerns. Previously, I thought the analyses had been carefully conducted and were of interest but the descriptions and interpretations of findings were confusing and could have been misleading to readers. The revised manuscript is very clear and an impressive presentation.

Reviewer #2: The author's have clarified all comments I raised. No further changes are needed.

**Have all data underlying the figures and results presented in the manuscript been provided?**

Reviewer #1: Yes

Reviewer #2: None

PLOS authors have the option to publish the peer review history of their article (what does this mean?). If published, this will include your full peer review and any attached files.

Reviewer #1: No

Reviewer #2: No

**Data Deposition**

http://datadryad.org/submit?journalID=pgenetics&manu=PGENETICS-D-23-00895R1

**Press Queries**

---

## [Editor Report · Acceptance letter]

5 Feb 2024

PGENETICS-D-23-00895R1 

Estimating the health impact of nicotine exposure by dissecting the effects of nicotine versus non-nicotine constituents of tobacco smoke: A multivariable Mendelian randomisation study 

Dear Dr Khouja, 

We are pleased to inform you that your manuscript entitled "Estimating the health impact of nicotine exposure by dissecting the effects of nicotine versus non-nicotine constituents of tobacco smoke: A multivariable Mendelian randomisation study" has been formally accepted for publication in PLOS Genetics! Your manuscript is now with our production department and you will be notified of the publication date in due course.

With kind regards,

Zsofi Zombor

PLOS Genetics

On behalf of:
